

# A revised parameterization for aerosol, cloud and precipitation pH for use in chemical forecasting systems (EQSAM4Clim-v12)

Swen Metzger[1], Samuel Rémy[2], Jason E Williams[3], Vincent Huijnen[3], and Johannes Flemming[4]

[1]ResearchConcepts Io GmbH, Freiburg, Germany
[2]HYGEOS, Lille, France
[3]R and D Weather and Climate modeling, Royal Netherlands Meteorological Institute, De Bilt, Netherlands
[4]European Centre for Medium Range Weather Forecasts, Reading, UK and Bonn, Germany

**Correspondence:** Swen Metzger (sm@researchconcepts.io)

**Abstract.** The Equilibrium Simplified Aerosol Model for Climate version 12 (EQSAM4Clim-v12) has recently been revised to provide an accurate and efficient method for calculating the acidity of atmospheric particles. EQSAM4Clim is based on an analytical concept that is not only sufficiently fast for numerical weather prediction (NWP) applications, but also free of numerical noise, which makes it attractive also for air quality forecasting. EQSAM4Clim allows the calculation of aerosol

composition based on the gas-liquid-solid and the reduced gas-liquid partitioning with the associated water uptake for both cases, and can therefore provide important information about the acidity of the aerosols. Here we provide a comprehensive description of the recent changes made to the aerosol acidity parameterization (referred to a version 12) which builds on the original EQSAM4Clim. We evaluate the pH improvements using a detailed box-model and compare against previous model calculations and both ground-based and aircraft observations from US and China covering different seasons and scenarios. We

show that, in most cases, the simulated pH is within reasonable agreement with the results of the E-AIM reference model and of satisfactory accuracy.

## 1 Introduction

In order to address the relevance of gas-aerosol partitioning and aerosol water for climate and air quality studies, the Equilibrium Simplified Aerosol Model (EQSAM) was developed as a compromise between numerical speed and accuracy (Metzger

et al., 2002). EQSAM has been widely used in many air quality and climate modelling systems worldwide (Metzger et al., 2018), including the IFS (Flemming et al., 2015) and the OpenIFS (Huijnen et al., 2022). Recently, the EQSAM version for Climate Applications (EQSAM4Clim) (Metzger et al., 2016a) has been implemented in IFS-COMPO (see the accompanying paper). In contrast to EQSAM, EQSAM4Clim is entirely based on a compound specific single-solute coefficient ($\nu_i$), which was introduced in Metzger et al. (2012) to accurately parameterise the single solution hygroscopic growth, considering the

Kelvin effect. This $\nu_i$-approach accounts for the water uptake of concentrated nanometre-sized particles up to dilute solutions, i.e. from the compounds relative humidity of deliquescence (RHD) up to supersaturation (Köhler theory). EQSAM4Clim extends the $\nu_i$-approach to multicomponent mixtures, including semi-volatile ammonium compounds and major crustal elements.





The advantage of EQSAM4Clim is that the entire gas–liquid–solid aerosol phase partitioning and water uptake, including major mineral cations, is solved analytically without iterations and thus computationally very efficient. This makes EQSAM4Clim suited not only for climate simulations, but also applicable to high resolution Numerical Weather Predictions (NWP) coupled with comprehensive atmospheric chemistry providing global values of particulate matter, as done in the Copernicus Atmosphere Monitoring Service (CAMS, Peuch et al. (2022); Rémy et al. (2022)) for example, using the ECMWF Integrated Forecasting System (IFS).

Previously, the use of EQSAM4Clim has undergone a rigorous assessment across different time scales, through a comparison with various observations and reference simulations on climate time scales using more than a decade of independent observations (e.g. Metzger et al. (2018)). Moreover, a comparison of simulated Aerosol Optical Depth (AOD) has been made against various satellite data at NWP time scales to validate the Polar Multi-sensor Aerosol properties (PMAp) AOD product version 2 AOD at an 1 hourly time resolution (Metzger et al., 2016b). EQSAM4Clim has been also used as part of air quality assessments through, e.g., the 2019 European Monitoring and Evaluation Programme (EMEP) report on transboundary particulate matter, photo-oxidants, acidifying and eutrophying components (Fagerli et al., 2019), and evaluated in the air quality modeling system CAMx over the continental US with 12 km grid resolution for winter and summer months (Koo et al., 2020). On all time scales, it was found that EQSAM4Clim accurately parameterises the gas/liquid/solid aerosol partitioning and associated aerosol water uptake sufficiently fast and free of numerical noise. This is due to its unique analytical structure, which makes it particularly also attractive for air quality assessments such as those provided by CAMS. Most recently, the latest version of EQSAM4Clim has been and implemented in IFS-COMPO as presented in Metzger et al. (2022) and Metzger et al. (2023). A more comprehensive evaluation of the performance on global pH values and resulting effects on Particulate matter in IFS-COMPO will be presented in two companion papers.

This Technical Note provides a description of the improved aerosol acidity parameterization applied in EQSAM4Clim-v12. We show an extensive validation against reference model calculations using Extended Aerosol Inorganics Model (E-AIM) as described in Wexler and Clegg (2002) and Friese and Ebel (2010), using the detailed case study on aerosol acidity provided by Pye et al. (2020).

## 2 Description of EQSAM4Clim-v12

The overall gas/liquid/solid partitioning and aerosol water uptake parameterization is the same as described and evaluated in Metzger et al. (2012) and Metzger et al. (2016a), with further evaluation being provided in Metzger et al. (2018). Here we limit the description to those new features added to previous versions.

### 2.1 General Features

A schematic of the various input parameters needed for use in EQSAM4Clim is shown in Figure 1, where chemical species from each phase type is given. EQSAM4Clim is based on a compound specific single-solute coefficient ($\nu_i$), which was introduced in Metzger et al. (2012) for single solute solutions and extended to multi-component mixtures by Metzger et al. (2016a) to

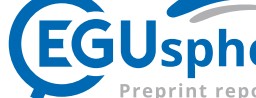

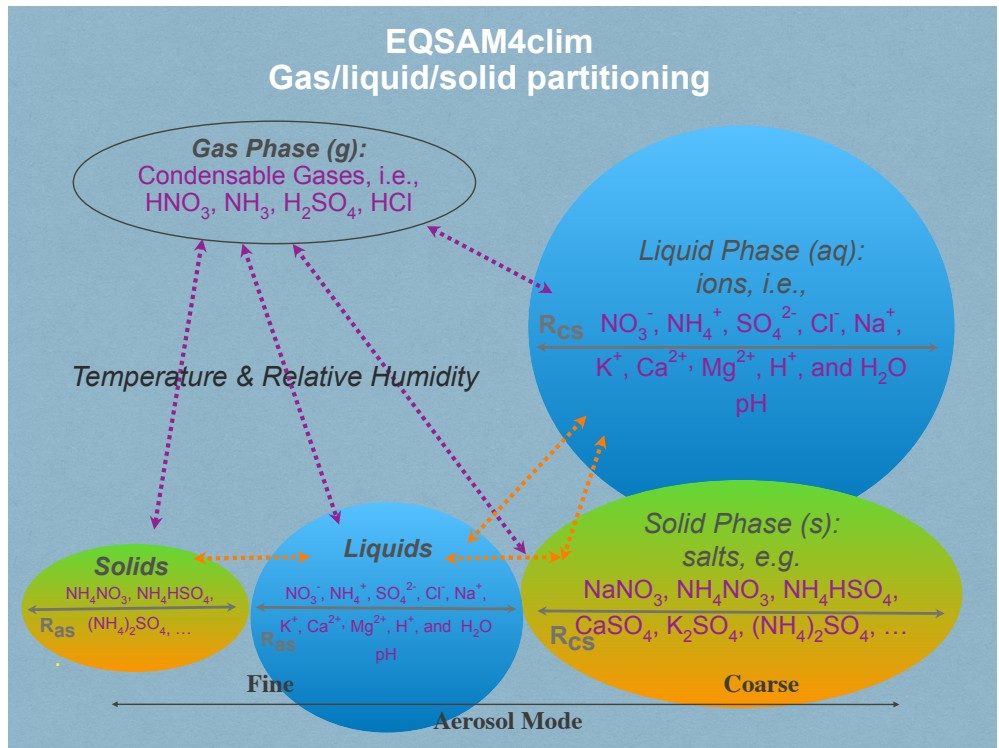

**Figure 1.** A schematic of the components included in EQSAM4Clim.

include semi-volatile ammonium ($NH_4^+$) compounds and major crustal elements. A feature of the $\nu_i$-approach is that the entire gas–liquid–solid aerosol phase partitioning and water uptake can be solved analytically without iterations, and hence without numerical noise.

EQSAM4Clim takes as input (i) the meteorological parameters air temperature (T) and relative humidity (RH), (ii) the aerosol precursor gases, i.e., major oxidation products of natural sources and anthropogenic air pollution represented by ammonia ($NH_3$), hydrochloric acid (HCl), nitric acid ($HNO_3$), sulphuric acid ($H_2SO_4$), and (iii) the ionic aerosol concentrations, i.e., lumped (both liquid and solid) anions, sulphate ($SO_4^{2-}$), bi-sulphate ($HSO_4^-$), nitrate ($NO_3^-$), chloride ($Cl^-$), and lumped (liquid+solid) cations, $NH_4^+$, sodium ($Na^+$), potassium ($K^+$), magnesium ($Mg^{2+}$) and calcium ($Ca^{2+}$).

The equilibrium aerosol composition and aerosol Associated Water mass (AW) is calculated by EQSAM4Clim through the neutralization of anions by cations, which yields numerous salt compounds, i.e., the sodium salts $Na_2SO_4$, $NaHSO_4$, $NaNO_3$, NaCl, the potassium salts $K_2SO_4$, $KHSO_4$, $KNO_3$, KCl, the ammonium salts $(NH_4)_2SO_4$, $NH_4HSO_4$, $NH_4NO_3$, $NH_4Cl$, the magnesium salts $MgSO_4$, $Mg(NO_3)_2$, $MgCl_2$, and the calcium salts $CaSO_4$, $Ca(NO_3)_2$, $CaCl_2$. All salt compounds (except $CaSO_4$) can partition between the liquid and solid aerosol phase, depending on T, RH, AW and the temperature-dependent Relative Humidities of Deliquescence of (a) single solute compound solutions (RHD) and (b) of mixed salt solutions (Metzger et al., 2016a).





Based on the RHD of the single solutes, the (mixed) solution liquid/solid partitioning is calculated, whereby all compounds for which the RH is below the RHD are assumed to be precipitated, such that a solid and liquid phase can coexist. The liquid-solid partitioning is strongly influenced by mineral cations and in turn largely determines the aerosol pH (Sect. 3).

EQSAM4Clim estimates the concentration of the hydronium ion ($H^+$) $[mol/m^3(air)]$ and, subsequently, the pH of the solution from electroneutrality ($Z^0$ $[mol/m^3(air)]$) after neutralization of all anions by all cations in the system (following

the neutralization reaction order given by Table 3 of Metzger et al. (2016a)). Note that the auto dissociation of $H_2O$ is taken into account, but currently no dissolution and dissociation of aerosol precursor gases such as sulphur dioxide ($SO_2$), nitric acid ($HNO_3$), hydrogen chloride ($HCl$), or ammonia ($NH_3$) is taken into account, as this is typically considered in the aqueous phase chemistry module of any global chemistry forecast model. The initial $H^{+,0}$ concentration $[mol/m^3(air)]$ after cation-anion neutralization is obtained from:

$$Z^0 = tAnions - tCations = \sum_i [Z^-]_i - \sum_j [Z^+]_j \tag{1}$$

$$[H^{+,0}] = Z^0 = 2[SO_4^{2-}] + [HSO_4^-] + [NO_3^-] + [Cl^-] - [K^+] - 2[Ca^{2+}] - 2[Mg^{2+}] - [Na^+] - [NH_4^+] \tag{2}$$

with tAnions and tCations (hereafter referred to as tCAT) representing the total of all anions and cations $[mol/m^3(air)]$, respectively and where $[H^{+,0}]$ denotes the initial hydronium ion concentration per volume air and which also depends on the auto-dissociation of $H_2O$ ($K_w$) $[mol^2/kg^2(water)]$. This is derived from Eq. (3) considering the temperature dependency as

widely assumed in equilibrium models.

$$K_w = 1.010 \times 10^{-14} \cdot \exp\left(-22.52 \cdot (\frac{T_0}{T} - 1) + 26.920 \cdot A_T\right) \qquad where \qquad A_T = \left(1 + \log(\frac{T_0}{T}) - \frac{T_0}{T}\right), \tag{3}$$

with $T_0 = 298K$.

## 2.2 Updates to the acidity component

### 2.2.1 Dependency of $H^+$ on the Chemical Domain

The neutralization equation does not correct for non-ideal solutions, such as described in Pye et al. (2020) and the references therein. For that purpose, with v12, we introduce for EQSAM4Clim a new factor XN, which is dependant on the aerosol composition, to correct the initial $[H^{+,0}]$ (Eq. 2). XN is obtained from:

$$XN = [X]/[Y] \tag{4}$$

with X denoting the sum of all anions noted above, while $Y = tNH_4$, i.e., the sum of $tNH_3$ and $tNH_4^+$. XN is applied without

further scaling factors for ranges of $XN < 0.9$ with ambient temperatures below 293K.

For cases outside this range, XN needs to be scaled by 10 and multiplied by the factor N given in Table 1, in order to account for chemical processes which are not resolved by the parameterizations (particularly concerning $HSO_4^-$ and free $H_2SO_4$).



**Table 1.** H$^+$ correction factors introduced with EQSAM4Clim-v12 for the chemical domains introduced in Metzger et al. (2016a).

| Domain | Characterization | Regime | | | | Correction factor N | Relation |
|--------|------------------|--------|---|---|---|---------------------|----------|
| D1 | CATION RICH | tCAT-tNH4 $\geq$ tSO$_4$ | | | | N=1 | XN |
| D2 | SO$_4$$^{2-}$ NEUTRAL | tCAT $\geq$ tSO$_4$ | | | | N=1 | XN |
| D3 | SO$_4$$^{2-}$ RICH | tCAT $\geq$ tHSO$_4$ | AND | tCAT $<$ tSO$_4$ | | N=1e1 | XN |
| D4 | SO$_4$$^{2-}$ VERY RICH | tCAT $\geq$ MIN | AND | tCAT $<$ tHSO$_4$ | | N=1e3 | – |

Following Table 2 of Metzger et al. (2016a), four chemical domains are considered to correct [H$^{+,0}$] obtained with Eq. (2).
No additional correction ($N = 1$) is needed for the neutral cases (D1-D2), i.e. where cations are in excess of total SO$_4$$^{2-}$, thus
preventing the formation of all HSO$_4$$^-$ salts (see Table 1 of Metzger et al. (2016a)). For the SO$_4$$^{2-}$ rich case (D3), XN and N
from Table 1 are multiplied, while for the SO$_4$$^{2-}$ very rich case (D4), only a constant correction factor (N) is applied to correct
Eq. (2), where tCAT denotes the sum of cations given in Eq. (2), tSO$_4$ is the sum of all SO$_4$$^{2-}$, including HSO$_4$$^-$ and H$_2$SO$_4$.

Additionally, we consider three cases for estimating the H$^+$ concentration, according to the possible solutions of Eq. (1), i.e.:

$$Z^* < 0 \quad | \quad [H^{+,*}] = \frac{\text{LWC}_{tot}}{10^{(7.0+\log(-Z^* \cdot \frac{10^4}{\text{LWC}_o \cdot \mu_s^o}))}} \cdot \mu_s^o \tag{5a}$$

$$Z^* = 0 \quad | \quad [H^{+,*}] = [H^{+,neutral}] \times 10^{-6}) \tag{5b}$$

$$Z^* > 0 \quad | \quad [H^{+,*}] = Z^* \times 10^{-6}) \tag{5c}$$

with LWC$_{tot}$ being the total Liquid Water Content [kg(H$_2$O)/m$^3$(air)] as defined below in Eqs. (9a-9d). LWC$_o$ = 1
[kg/m$^3$(air)] and $\mu_s^o = 1$ [mol/kg(H$_2$O), a reference solution and reference molality, respectively, to match units (Metzger
et al., 2012, 2016a; Pye et al., 2020). Z$^*$ is given by Eq. (6) and denotes the sum of our initial hydrogen concentration [H$^{+,0}$]
and [H$^{+,neutral}$], an effective hydrogen concentration in a neutral solution (pH=7), which is given by Eq. (7), but empirically
derived for our parameterization:

$$Z^* = [H^{+,neutral}] + [H^{+,0}] \tag{6}$$

$$[H^{+,neutral}] = \frac{B \cdot \text{LWC}_o \cdot \text{K}_w^{0.5}}{(1.0 - \text{RH}^2)} \tag{7}$$

with K$_w$ from Eq. (3), a constant $B = 1/(\mu_s^o \cdot m_w) = 55.51$ [$-$], the molar mass of water, m$_w$ [kg/mol], and the reference
molality $\mu_s^o$ [mol/kg(H$_2$O)] and a reference solution LWC$_o$ = 1 [kg/m$^3$(air)] to match units; RH denotes the fractional relative
humidity [0-1].

Finally, the H$^+$ concentration of a given solution is obtained from:

$$[H^+] = [H^{+,*}] \cdot XN \tag{8}$$



### 2.2.2 Dependency of pH on the Liquid Water Content

For EQSAM4Clim-v12, five different pH values can be computed from the revised H$^+$ [mol/m$^3$(air)] computation (Sect. 2.2.1) for diagnostic output. Therefore, EQSAM4Clim-v12 allows the differentiation of the various LWC [kg(H$_2$O)/m$^3$(air)] values associated with different type of atmospheric aerosols, haze/fog, or cloud droplets contained in the troposphere as defined in Eqs. (9a-9e):

$$pH_{equil} = -\log_{10}\left(\frac{[H^+]}{LWC_{equil}} \cdot \frac{1}{\mu_s^o}\right) \tag{9a}$$


$$pH_{noneq} = -\log_{10}\left(\frac{[H^+]}{LWC_{noneq}} \cdot \frac{1}{\mu_s^o}\right) \tag{9b}$$

$$pH_{cloud} = -\log_{10}\left(\frac{[H^+]}{LWC_{cloud}} \cdot \frac{1}{\mu_s^o}\right) \tag{9c}$$

$$pH_{precip} = -\log_{10}\left(\frac{[H^+]}{LWC_{precip}} \cdot \frac{1}{\mu_s^o}\right) \tag{9d}$$

$$pH_{total} = -\log_{10}\left(\frac{[H^+]}{LWC_{total}} \cdot \frac{1}{\mu_s^o}\right) \tag{9e}$$

Here, (i) LWC$_{equil}$ [kg(H$_2$O)/m$^3$(air)] denotes the equilibrium water content calculated within EQSAM4Clim (from Eq.(22)
in Metzger et al. (2016a)), (ii) LWC$_{noneq}$ is the aerosol liquid water content associated with aerosol species not considered in the equilibrium computations of EQSAM4Clim (e.g., from chemical aging of pre-existing organic or black carbon particles as used e.g. in Metzger et al. (2016b) and Metzger et al. (2018)), (iii) LWC$_{cloud}$ denotes the cloud liquid water content, (iv) LWC$_{precip}$ denotes the liquid water content of a given precipitation flux and finally (v) LWC$_{total}$ denotes the sum of LWC of Eqs. (9a-9d), where $\mu_s^o = 1$ [mol/kg(H$_2$O)] is the reference molality to match units. The pH values of Eqs. (9b-9e) are an optional output
feature and requires the corresponding input to EQSAM4Clim-v12 (e.g., in any 3-D application these are provided by the forecasting model). It is important to note that all pH and H$^+$ values are only for diagnostic output, as these values are not used within EQSAM4Clim.

In contrast to other aerosol equilibrium models such as E-AIM, EQSAM4Clim has a fully analytical structure which does not depend on the pH. However, accounting for the different pH values is important for air quality and climate applications,
because of the influence of solution pH on aqueous phase chemistry in terms of SO$_4$$^{2-}$ production and the subsequent deposition processes. Also note that in the case where an accurate pH calculation is needed, reference calculations from E-AIM should be considered instead. With EQSAM4Clim-v12 we try to find a compromise between computational speed and accuracy, so not always the pH parameterization might be applicable, although we show EQSAM4Clim-v12 performs well over a wide range of atmospheric conditions (see Sect. 3).



## 3 Results and Evaluation

In this section, we present comparisons of the revised pH parameterization of EQSAM4Clim-v12 against a wide range of data from different measurement sites and field campaigns spanning different locations for different years, where this data has been used in the comprehensive pH review study of Pye et al. (2020). In Pye et al. (2020) five distinct cases were defined and used to evaluate the simulated pH of various thermodynamic models applied in large-scale models. We therefore select the same observational data for input to the box model calculations and also use the E-AIM model output as a reference to evaluate the revised pH parameterization. It should be noted that E-AIM is much too computationally expensive to be applied in a large-scale atmospheric model. Details of the five cases used are provided below (see also Table 5 of Pye et al. (2020) and the references therein), i.e., here sorted by complexity of the chemical system with respect to the aerosol composition:

- Europe: Cabauw Experimental Site for Atmospheric Research (CESAR), Cabauw (51.970° N, 4.926° E), The Netherlands, 2 May 2012–4 Jun 2013, 2646 data points, composition including:
  $Mg^{2+}$–$Ca^{2+}$–$K^+$–$Na^+$–HCl/$Cl^-$–$NH_3$/$NH_4^+$–$HNO_3$/$NO_3^-$–$H_2SO_4$/$SO_4^{2-}$/$HSO_4^-$–$H_2O$

- Asia: Measurement site Tianjin (39.7° N, 117.1° E), China, 9–22 Aug 2015, 241 data points, composition including:
  $Mg^{2+}$–$Ca^{2+}$–$K^+$–$Na^+$–HCl/$Cl^-$–$NH_3$/$NH_4^+$–$HNO_3$/$NO_3^-$–$H_2SO_4$/$SO_4^{2-}$/$HSO_4^-$–$H_2O$

- SE US: Southern Oxidant and Aerosol Study (SOAS) campaign, Centreville (32,9° N, 87,1° W), US, 6 Jun–14 Jul 2013, 787 data points, reduced composition (no mineral cations):
  $Na^+$–HCl/$Cl^-$–$NH_3$/$NH_4^+$–$HNO_3$/$NO_3^-$–$H_2SO_4$/$SO_4^{2-}$/$HSO_4^-$–$H_2O$

- SW US: California Nexus (CalNex) campaign, Pasadena (34,15° N, 118,1° W), CA, USA, 17 May–15 Jun 2010, 493 data points, reduced composition (no mineral cations, and no sodium):
  HCl/$Cl^-$–$NH_3$/$NH_4^+$–$HNO_3$/$NO_3^-$–$H_2SO_4$/$SO_4^{2-}$/$HSO_4^-$–$H_2O$

- NE US: Wintertime Investigation of Transport, Emissions, and Reactivity (WINTER) campaign, Eastern US aloft, 3 Feb 2015, 3613 data points, reduced composition (no mineral cations, and no sodium):
  HCl/$Cl^-$–$NH_3$/$NH_4^+$–$HNO_3$/$NO_3^-$–$H_2SO_4$/$SO_4^{2-}$/$HSO_4^-$–$H_2O$

A total of more than 7700 data points are available for evaluation from these campaigns covering a wide range of RH (above 20 %RH) and T ($\approx$ 250 to 310 K). Moreover, relevant input is provided for assessing the performance over a complete year (Cabauw), summertime (e.g. SOAS) and wintertime (WINTER). Note that only the correction factors needed for the revised $H^+$ computations (shown in Table 1) have been iteratively derived by comparing the diagnostic pH output of EQSAM4Clim-v12 with the reference pH computations of E-AIM for these five cases (using error minimizing on the log-scale), while the water uptake calculation is identical to that described in Metzger et al. (2016a).





## 3.1 Aerosol pH

To evaluate the revised pH parameterization of EQSAM4Clim-v12, we compare the resulting values against the output of
the E-AIM reference model for the five field campaign cases as discussed in Pye et al. (2020). In Figure 2 the results of
EQSAM4Clim-v12 and E-AIM are shown and compared to measurements at the Cabauw Experimental Site for Atmospheric
Research (CESAR, Guo et al. (2018b)), for the period of May 2012 - June 2013. The AW and aerosol pH are shown together
with the corresponding T and the RH data, which are used as meteorological input to this box modelling study together, with

the ion concentrations taken from the reference study. There is a seasonal cycle in T throughout the year, with a typical range in
RH of between 60-90% with the measurement station being representative of a polluted rural site. For pH, the spread simulated
by EQSAM4Clim-v12 ranges from pH 1.8-4.5 which is close to that from E-AIM, whose minimum pH is around 2.0 in a range
between 2.0-5.0 i.e. less acidic. Also the aerosol water predictions of both models compare well throughout all seasons, with
only a few noticeable exceptions in spring 2013 (around step 2000). Note the input data here is unfiltered and may include a

few outliers that are not valid. Also note that the $pH_F$ refers to the free-$H^+$ approximation of pH which is only included for
completeness, but not further used and discussed here (we refer the interested reader to Pye et al. (2020)).

   Figures 3–5 show similar comparisons for summertime for the Tianjin (Shi et al. (2019)), SOAS (Alabama Forest, US; Guo
et al. (2015)) and CalNex (Pasadena, US; Guo et al. (2018a)) campaigns, representing both urban and forest scenarios between
the years 2010 and 2015. The range in T for these campaigns is typically limited to between 290-310K, with distinct signatures

of diurnal variability in the RH. This results in similar variability in the AW content, with the pH range in SOAS (-1.0 to 2.0)
being order of magnitude more acidic than either CalNex (1.0-5.0) or Tianjin (2.0-5.0). Again the spread in the pH values from
EQSAM4Clim-v12 is here similar to that of the E-AIM reference model and only a bit wider for the CalNex case compared to
the Cabauw case, due to limitations of the bi-sulfate/sulfate partitioning of the EQSAM4Clim version.

   Finally for wintertime under polluted conditions we use the data from one flight taken as part of the WINTER flight campaign

(US East Coast;Guo et al. (2016). Here EQSAM4Clim-v12 is tested for lower temperatures across a wide range of RH values
at various altitudes with high values of sea-salt. Figure 6 shows comparisons using data from the flight taken on 3rd February
2015. Both EQSAM4Clim-v12 pH and E-AIM simulate low pH values estimates, with similar variability and correlated well
with respect to pH values < 0.0.

   Figure 7 shows a comparison of the EQSAM4Clim-v12 pH results of the previous version 10 (left) (used e.g. in (Fagerli

et al., 2019) and (Koo et al., 2020)) and the current version 12 (right) versus the pH results of E-AIM for all five cases. Clearly,
the pH results of EQSAM4Clim-v12 pH are closer to E-AIM compared to the v10, now more closely following the one-by-one
line for a wide range of atmospheric conditions, although some scatter still remains. Note that this scatter is acceptable for the
EQSAM4Clim parameterization concept. A more explicit treatment of the phase partitioning will be subject of a follow-up
study.





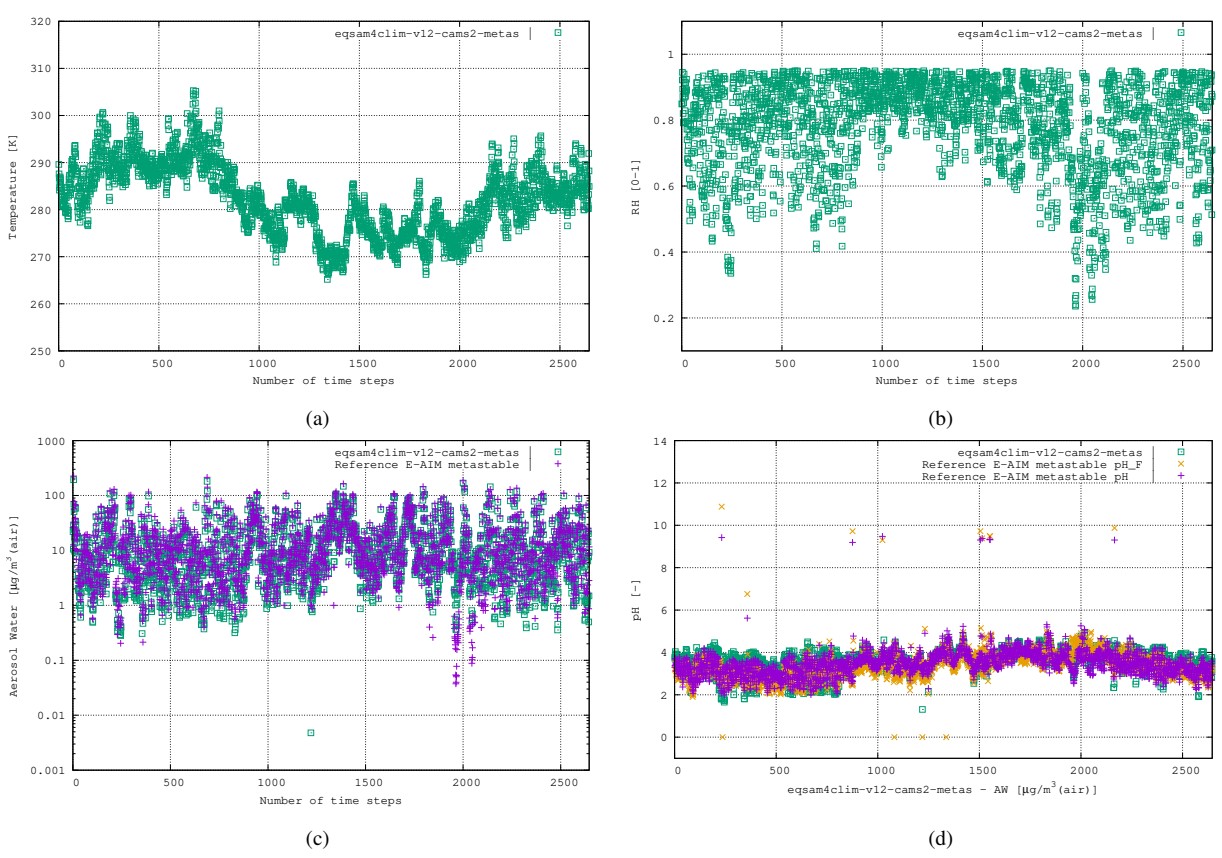

**Figure 2.** First case study: CESAR site, Cabauw, The Netherlands, 2 May 2012–4 Jun 2013, 2646 data points. The results of EQSAM4Clim-v12 (green, squares) in comparison with E-AIM (pink, cross) using the data provided by Pye et al. (2020). Both models use the T [K] (panel a) and RH [0-1] (panel b) together with the lumped ion concentrations [$\mu g/m^3$(air)] of $Mg^{2+}$, $Ca^{2+}$, $K^+$, $Na^+$, $HCl+Cl^-$, $NH_3+NH_4^+$, $HNO_3+NO_3^-$, $H_2SO_4+SO_4^{2-}+HSO_4^-$ as input to calculate the aerosol water mass, $H_2O$ [$\mu g/m^3$(air)] (panel c) and aerosol pH [-] (panel d), assuming the metastable aerosol phase (no solid/liquid, only gas/liquid partitioning aerosol partitioning). Additionally, the E-AIM output for the free pH ($pH_F$, orange X) is included (see Pye et al. (2020)).



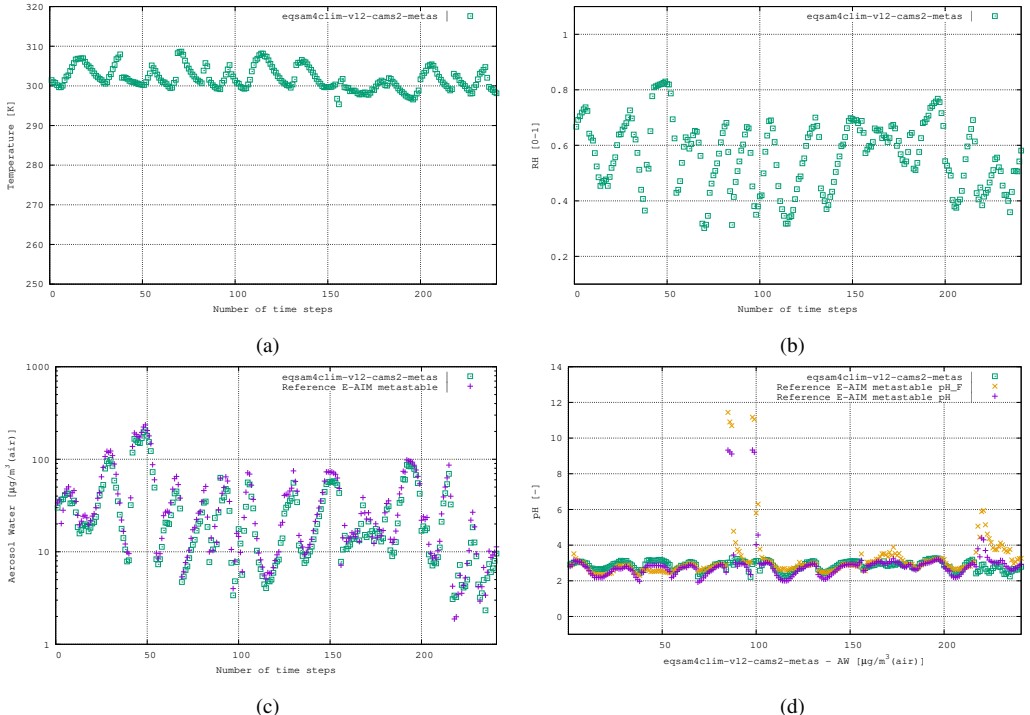

**Figure 3.** Second case study: Tianjin, China, 9–22 Aug 2015, 241 data points, using the aerosol system: $Mg^{2+}$, $Ca^{2+}$, $K^+$, $Na^+$, $HCl+Cl^-$, $NH_3+NH_4^+$, $HNO_3+NO_3^-$, $H_2SO_4+SO_4^{2-}+HSO_4^-$.

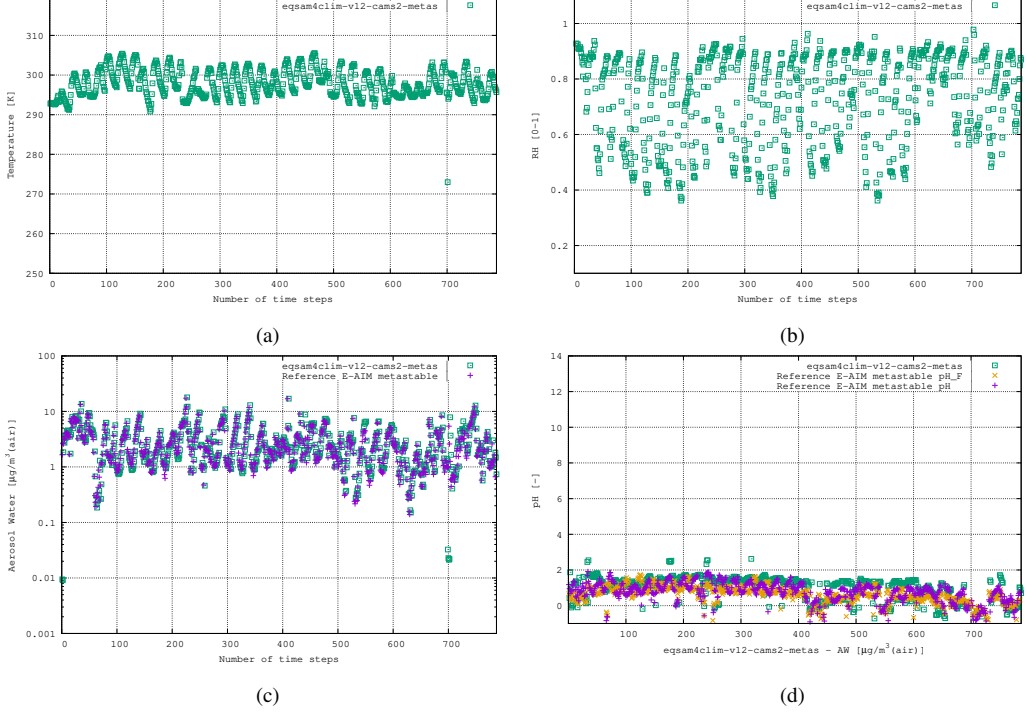

**Figure 4.** Third case study: SOAS campaign, Centreville, US, 6 Jun–14 Jul 2013, 787 data points, using the aerosol system: $Na^+$, $HCl+Cl^-$, $NH_3+NH_4^+$, $HNO_3+NO_3^-$, $H_2SO_4+SO_4^{2-}+HSO_4^-$.





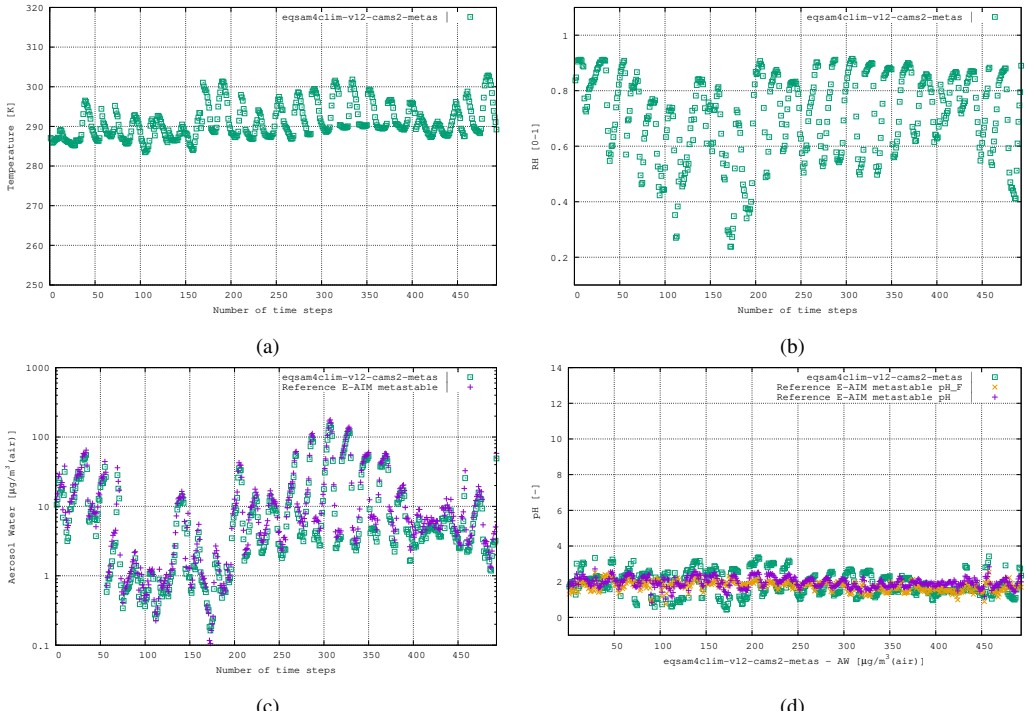

**Figure 5.** Fourth case study: CalNex campaign, Pasadena, CA, USA, 17 May–15 Jun 2010, 493 data points, using the aerosol system: $HCl+Cl^-$, $NH_3+NH_4^+$, $HNO_3+NO_3^-$, $H_2SO_4+SO_4^{2-}+HSO_4^-$.

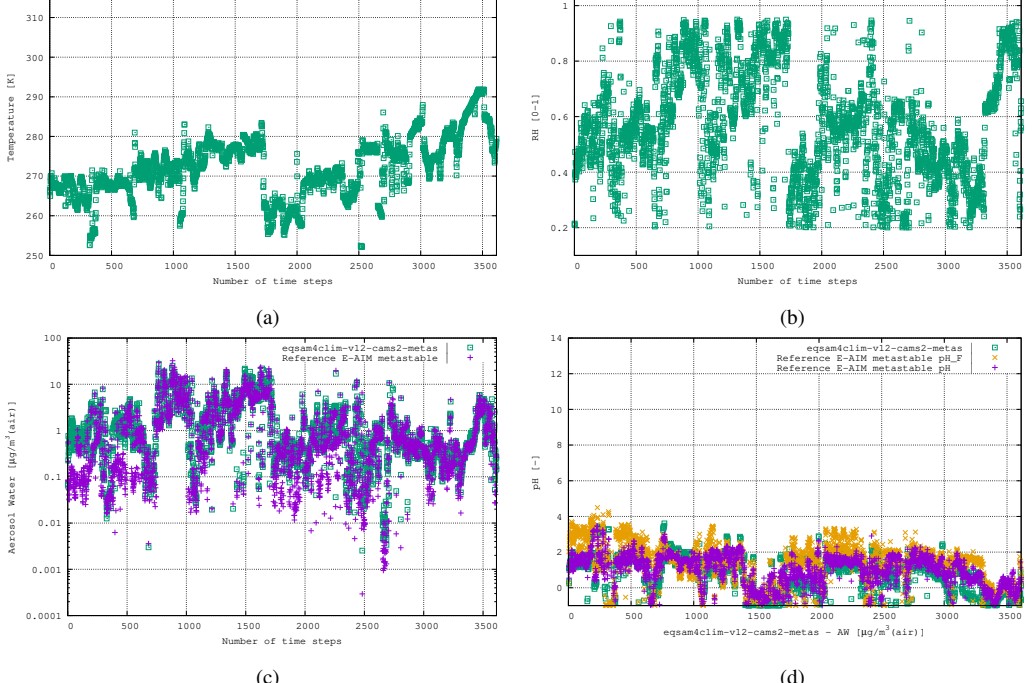

**Figure 6.** Fifth case study: WINTER campaign, Eastern US aloft, 3 Feb 2015, 3613 data points, using the aerosol system: $HCl+Cl^-$, $NH_3+NH_4^+$, $HNO_3+NO_3^-$, $H_2SO_4+SO_4^{2-}+HSO_4^-$.





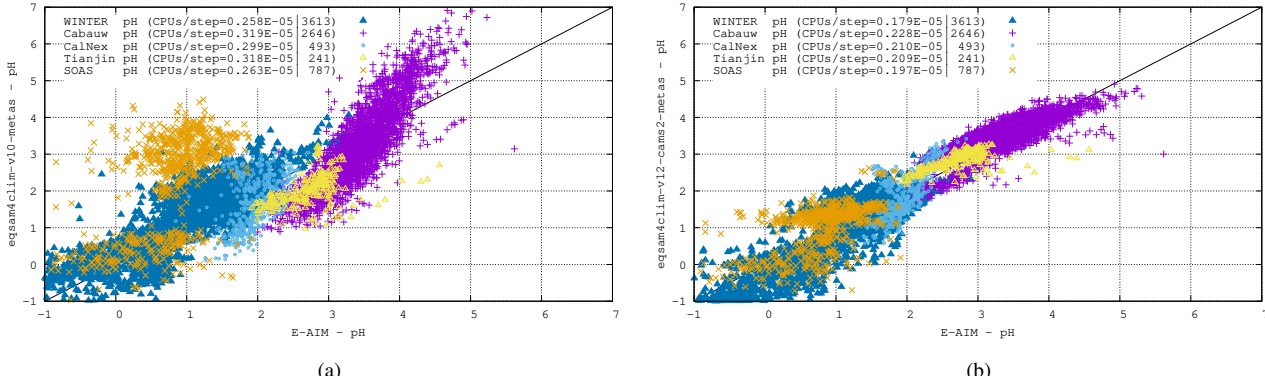

**Figure 7.** Comparison of the EQSAM4Clim pH results of v10 (panel a) and v12 (panel b) versus the pH results of E-AIM for all five cases. The CPU consumption per step is included for each case. Chip: Apple M1 Ultra; Memory: 128 GB; llvm-11/flang compiler with O3.

**Table 2.** Statistical metrics for the pH results of EQSAM4Clim pH results of v12 (left) and E-AIM (right column).

| Campaign | Data Min | | Data Max | | Data Mean | | Std.-Dev. | | Bias | Corr. | Count |
|---|---|---|---|---|---|---|---|---|---|---|---|
| Cabauw | 1.308 | 2.000 | 4.906 | 5.617 | 3.575 | 3.448 | 0.493 | 0.521 | 0.127 | 0.829 | 2646 |
| Tianjin | 2.171 | 1.921 | 3.270 | 4.565 | 2.838 | 2.743 | 0.250 | 0.389 | 0.095 | 0.595 | 241 |
| SOAS | -0.719 | -0.908 | 2.622 | 1.909 | 0.988 | 0.763 | 0.640 | 0.534 | 0.225 | 0.564 | 787 |
| CalNex | 0.428 | 0.844 | 3.418 | 2.836 | 1.907 | 1.957 | 0.649 | 0.288 | -0.05 | 0.731 | 493 |
| WINTER | -1.000 | -0.996 | 3.609 | 3.472 | 0.934 | 1.019 | 0.936 | 0.831 | -0.085 | 0.874 | 3613 |

To scrutinize the sensitivity and computational costs of these results, the results of two EQSAM4Clim versions including the CPU consumption per step are given in the panels of Figure 7 for each case. Comparing the two EQSAM4Clim versions (left and right panel) shows (a) that the pH results differ mostly for the Cabauw, Tianjin and SOAS campaigns, which represent different aerosol compositions and neutralization levels as defined by where the measurement campaign took place. While the Cabauw and Tianjin campaigns represent the most complex aerosol system with $SO_4^{2-}$ being fully neutralized (Sect. 3), where both locations are affected by anthropogenic precursors which undergo gas/aerosol partitioning. Conversely, data from the SOAS, Calnex and the WINTER campaigns represent cases where $SO_4^{2-}$ is not fully neutralized. Especially, the measurements from CalNex and the flight during the WINTER campaign represent often highly acidic cases.

Additionally, comparing cases shows that (b) the variability in the observed pH ranges across campaigns exceeds the variability in pH simulated by the different modeling code versions. For instance, the pH values are for the WINTER campaign generally much lower compared to e.g., the Cabauw campaign, which shows throughout all results the highest pH values, reflecting the predominance of cations in the aerosol system for the Cabauw case.

Table 2 summarizes the key metrics and shows for each campaign the minimum and maximum pH value, together with the data mean and standard deviation for EQSAM4Clim (v12) and E-AIM, as well as the correlation of both. While the data





mean is with a variation of less than 0.25 pH units generally satisfactorily close for all campaigns, the correlation coefficient is
only above 0.7 for the Cabauw, CalNex and WINTER campaign. Tianjin, which represents besides Cabauw the most complex
aerosol system, shows a slightly lower correlation coefficient of 0.6, while SOAS is with a value of 0.56 at the lower end, due
to the influence of sulfate/bi-sulfate partitioning. Bi-sulfates are not always captured in the gas/liquid partitioning compared
to cases which include semi-volatile compounds (Cabauw, Tianjin, WINTER). Also note that the correlation coefficient is
strongly influenced by the number of data points, such that the WINTER and Cabauw cases are statistically more significant.

This complexity of the Cabauw data is also reflected in the highest computing consumption per step (where CPU/step values
are given in the legend within each panel of Figure 7), while the WINTER campaign represents the least complex system (no
cations and low temperatures) and, therefore, requires also the least CPU time. Note that there is some uncertainty in these
numbers due to the load imbalance of the system ($\leq 1\%$), while the CPU consumption for EQSAM4Clim-v10 is higher due to
the fact that double precision is used. For EQSAM4Clim-v12, the choice of precision is optional and single precision is used
throughout this work, since this alone can speed up the computations of up to 50% for these run-time optimized cases.

## 3.2 Application to IFS

Figure 8 extends Figure 2 by showing an example of application and implementation of EQSAM4Clim-v12 into a comprehen-
sive high resolution atmospheric chemistry NWP forecatsing system, with the chosen model being IFS-COMPO (Peuch et al.,
2022). We use a version similar to that described in Rémy et al. (2022), where the gaseous precursors such as $HNO_3$ are derived
using the chemistry scheme given in Williams et al. (2022). The implementation provides global pH values, whose impact on
PM2.5 will be evaluated in a companion paper. Here we only compare the new optional IFS pH results using a 3 hourly output
frequency with those from the EQSAM4Clim-v12 and E-AIM box models at an hourly frequency for the Cabauw case as an
example. Note that only a qualitative comparison is possible due to the cumulative effects of the different time averages, the
difference in resolution (with IFS-COMPO being ran at $\approx 25$km scale) and in that the IFS-COMPO pH results are representa-
tive for 2019, while the box models show the results for the year 2012/2013. Nevertheless, overall the seasonal trend of the pH
values is captured rather well. During summertime (June - September) the IFS-COMPO pH values are on average less acidic
than those calculated in the box models, while for the autumn and winter months the agreement is on average very good. This
evaluation furthermore illustrates that uncertainties to the input quantities (aerosol composition, relative humidity, temperature)
dominate the overall uncertainty, indicating that EQSAM4Clim is fit for purpose.

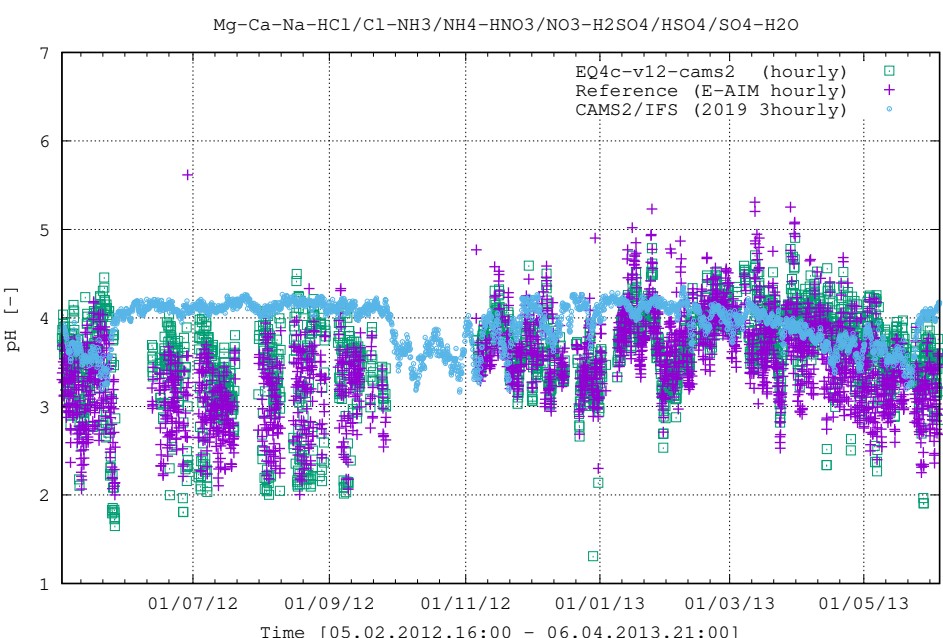

**Figure 8.** Figure 2 extended to include IFS (3 hourly averages) in comparison with EQSAM4Clim-v12 and E-AIM box model results.



## 4 Conclusions

In this technical note we have provided a description of the revised EQSAM4Clim-v12 pH parameterization developed for use in regional and global chemical forecasting systems. Using a range of diverse case studies we have performed box model calculations with the results being compared against those from the E-AIM reference model calculations, covering a range of seasons and scenarios ranging from forest measurements to maritime seaboard measurements. Generally, the pH values are mostly within the range given by E-AIM and now more closely following the one-by-one line for a wide range of atmospheric conditions compared to the previous EQSAM4Clim version (v10). Although some deviations of the EQSAM4Clim-v12 pH estimates are noted, the scatter is acceptable for the EQSAM4Clim parameterization concept. The case studies reveal that the pH results of the revised parameterization provide satisfactory representation for the most complex aerosol cases, i.e., the sulfate neutral conditions which are characterized by the gas/aerosol partitioning of semi-volatile ammonium compounds. Additionally, EQSAM4Clim has a low cost with respect to the overall CPU consumption across aerosols with a range of composition complexity (as seen across different campaigns). This provides confidence that the revised pH parameterization is suitable for applications to large-scale chemical forecasting systems where near-real time results are mandatory.

*Code availability.* The current version of model is available from GitHub: https://github.com/rc-io/eqsam under the licence CC-BY-SA-4.0. The exact version of the model used to produce the results used in this paper is archived on Zenodo (Metzger, 2023), as are input data and scripts to run the model and produce the plots for all the simulations presented in this paper (Metzger, 2023).

*Author contributions.* SM drafted the paper and developed and implemented the revised pH formulation of EQSAM4Clim-v12 into IFS; SR, VH, SM, JW and JF maintain and carry out general aerosol and trace gas developments on IFS-COMPO; all contributed to drafting and revising this article.

*Competing interests.* At least one of the (co-)authors is a member of the editorial board of Geoscientific Model Development.

*Acknowledgements.* This work is supported by the Copernicus Atmospheric Monitoring Services (CAMS) program managed by ECMWF on behalf of the European Commission. We acknowledge the Pye et al. (2020) study for providing the data of the box modelling study.





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
