# Peer review of "A computationally efficient parameterization of aerosol, cloud and precipitation pH for application at global and regional scale (EQSAM4Clim-v12)"

_EGUsphere, 2023_

## Author Comment (AC1)

**Authors' Response to comments of Anonymous Referee #1**

**General Comments**

*The paper presents another important evaluation for the efficient thermodynamic and chemical aerosol model EQSAM including some additional information on parameterizations and an application. This should be reflected more clearly in the title. EQSAM can be used as module in global and regional models or forecasting systems which should be mentioned explicitly in the introduction. The manuscript might be published after minor revision.*

We thank the anonymous reviewer for her/his positive response and appreciate the thorough review and constructive feedback. We have addressed all comments to enhance the clarity and quality of our manuscript. Our point-by-point response (in black) addresses all comments (in blue).

Title Clarification: We acknowledge your suggestion regarding reflecting the importance of the evaluation of the EQSAM model more explicitly in the title. We have revised the title to better emphasize the significance of the evaluation and its application in global and regional models or forecasting systems. New title: "*A computationally efficient parameterization of aerosol, cloud and precipitation pH for application at global and regional scale (EQSAM4Clim-v12)*".

Introduction Clarity: Thank you for highlighting the importance of explicitly mentioning EQSAM's applicability as a module in global and regional models or forecasting systems in the introduction.

We have revised the introduction accordingly at line 24ff: "*This makes EQSAM4Clim suited not only for climate simulations, but also* applicable to *air quality applications at the regional and global scale and ideal for high resolution Numerical Weather Predictions (NWP) …*"

**Specific Comments**

*Line 96: This sentence has to be improved for clarity. Does this mean a total factor of 100 in case D3? The use of 'N' together with 'XN' is also confusing, better use another letter (or 'Xn').*

Line 96: Yes, this means a total factor of 100 in case D3. We'll consider using 'YN' instead of 'N' and we have revised the sentence for clarity to:
"For cases outside this range ($XN \geq 0.9$ or $T \geq 293K$), XN needs to be scaled by 10 and multiplied by the factor YN given in Table 1, …"

*Line 159: "including E-AIM" should be inserted after "models" for better understanding the following.*

Line 159: We have inserted "including E-AIM" after "models" for better comprehension. The revised sentence reads:
"In Pye et al. (2020) five distinct cases were defined and used to evaluate the simulated pH of various thermodynamic models applied in large-scale models, including E-AIM."

*Lines 167-177: This can be expressed in a shorter way.*
Lines 167-177: We have condensed it to two lines (instead of three) for each bullet by replacing "2646 data points" by "N=2646", "The Netherlands" by "NL", "China" by "CN", and by deleting "composition including" and "reduced composition" for each bullet point.

*Line 234: Refer also to Fig 7b and say that V10 is worse here.*

Line 234: We'll refer to Fig 7b and specify that V10 performs worse in this scenario:
"Bi-sulfates are not always captured in the gas/liquid partitioning compared to cases which include semi-volatile compounds (Cabauw, Tianjin, WINTER), but v12 still outperforms v10 (comparing Fig. 7a and b)."

*Technical corrections*

*Line 10: Define acronym, line 44 is too late.*
Line 10: Changed to: … Extended Aerosol Inorganics Model (E-AIM) …
*Line 13: The acronym is already defined in abstract.*
Line 13: Changed to: EQSAM was developed …
*Line 16: Define acronym IFS, line 28 too late.*
Line 16: Changed to: … ECMWF's Integrated Forecasting System (IFS) …
*Line 25: Acronym already defined in abstract.*
Line 25: Changed to: … NWP …

*Line 17: Has the accompanying paper a reference? Or the references in lines 27, 243f? Define acronym.*
Line 17: We have added the missing references of the the accompanying paper Rémy et al. (2024) (https://egusphere.copernicus.org/preprints/2024/egusphere-2023-3072) and we have defined the IFS-COMPO acronym so that the sentence now reads:

"… implemented in the Integrated Forecasting System (IFS), with extensions to represent aerosols, trace and greenhouses gases, being called "IFS-COMPO" (also previously known as "C-IFS", Flemming et al. (2015)); see the accompanying paper (Rémy et al. (2024))."

*Line 74: Define also Z\* and H+\* here (missing for Eqn.5).*
Line 74: We have extended the sentence to include: "… (following the neutralization reaction order given by Table 3 of Metzger et al. (2016a)),  by using the effective hydrogen concentrations $H^{+,*}$ and $Z^*$ that are derived from Eqs. (1–8)."

*Line 84: Parentheses messed up.*
Line 84: Changed to: … "auto-dissociation of water $K_w$ [$mol^2$ /$kg^2$ ($H_2O$)]".

*Line 91: Typo*
Line 91: Changed to: "depends".

*Lines 117 and 144: Remove the repetitions of the definitions in line 109f.*
Line 117, 144: We have removed the repetitions of the definitions.

*Line 166ff: This punctuation is confusing, use ',' instead of '-' between species and '+' instead of '/' as in caption of Fig. 2.*
Line 166ff: We have changed the punctuation accordingly.

*Figure 2 to 6: It would be better to use actual time as abscissa than the data point number (as Fig. 8). The additional title near the abscissa of panel d should be removed and might appear in caption.*
Figure 2-6: We prefer not to consider using the actual time as the abscissa for Figures 2 to 6, since the data gap is too large which will reduce and not enhance clarity. The gaps can be seen in Figure 8 for the Cabauw case, which shows the data too condensed to be clear for most of the other cases.

*Figure 8: Don't mess up different notations for time at abscissa (dd/mm/yy and mm.dd.yyyy.hh).*
Figure 8: We have changed the time notation for Figure 8.

---

## Author Comment (AC2)

**Authors' Response to comments of Anonymous Referee #2**

**Summary**

*The Technical Note presents an updated version of EQSAM, i.e. EQSAM4Clim-v12, which has been improved to simulate a more accurate pH of aerosol associated water (AW), and also in cloud water and precipitation. Accurate, numerically robust and efficient schemes to calculate the gas/aerosol partitioning, aerosol water and pH are indeed in large demand by regional and global chemical transport models (CTM) for both air quality assessments and forecasts. My recommendation is that the paper can be published in EGUsphere after minor revisions.*

We thank the anonymous reviewer for her/his positive response and appreciate the thorough review and very detailed feedback. We have addressed all comments to enhance the clarity and quality of our manuscript. Our point-by-point response (in black) addresses all comments (in blue).

**General Comments**

*I do not understand why it's stressed that the EQSAM4Clim-v12 is primarily meant for "chemical forecasting systems". It can indeed be implemented in any CTM used for Chemical Weather prediction and Air quality simulations, but also in climate models.*

We appreciate the clarification regarding the applicability of EQSAM4Clim-v12 beyond chemical forecasting systems. While the model is indeed versatile and can be implemented in various modeling frameworks, our emphasis on its use in chemical forecasting systems stems from its primary focus on providing predictions of atmospheric composition and pH levels. Nevertheless, we have changed the title and introduction to be more general.

New title: "*A computationally efficient parameterization of aerosol, cloud and precipitation pH for application at global and regional scale (EQSAM4Clim-v12)*".

Introduction changed at line 24ff: "*This makes EQSAM4Clim suited not only for climate simulations, but also* applicable to *air quality applications at the regional and global scale and ideal for high resolution Numerical Weather Predictions (NWP) …*"

*Then, I'd recommend to include in the relevant sections:*
*- brief explanation of the need/importance of modelling aerosol water pH*

Aerosol water pH plays a crucial role in the atmospheric chemistry of aerosols. It influences the aqueous uptake of gases such as sulfur dioxide or ammonia, which are key components affecting atmospheric composition and air quality. The acidity or alkalinity of aerosol water affects chemical reactions, including the formation and transformation of aerosol particles, as well as their interactions with other atmospheric constituents. Consequently, accurate modeling of aerosol water pH is essential for understanding and predicting processes related to atmospheric composition, aerosol-cloud interactions, and ultimately, their impact on climate and human health.

We therefore have added a short sentence about its impact at line 151:
"*… processes. The pH of aerosols controls their impact on climate and human health.*"

*- more clear outline of the improvements introduced with respect to the previous version and what particular processes and results (in what particular cases) they improved (was it only the factor XN, line 91?)*

We appreciate the request for a clearer outline of the improvements introduced in EQSAM4Clim-v12 compared to its previous version. Notably, EQSAM4Clim-v12 introduces a refined parameterization that separates aerosol, cloud, and precipitation pH, addressing a limitation of previous versions. This enhancement allows for more accurate representation of pH variations across different atmospheric conditions. The role of the XN correction factors can have a strong local impact in improving model performance as shown with Fig 7.

In the revised manuscript, we have added the following sentences at line 214:
"… study. Also note that both versions only differ by Eqs. (1-9e) with the results shown being sensitive to the Eq. (8) and the correction factors given in Table 1. Finally, note that what is most important for 3D applications is the fact that version 12 introduces a refined parameterization that separates the pH of aerosol, cloud and precipitation and addresses a limitation of previous versions through Eqs. (9a-9e). For a in-depth analysis we refer to the accompanying study of Rémy et al. (2024)."

*- what tests have been performed to check that the scheme is "free for numerical noise" (lines 4, 38, 57).*

We appreciate the inquiry regarding the numerical stability of the EQSAM4Clim-v12 scheme. As an analytical code, EQSAM inherently yields numerically stable results without introducing uncertainty through incomplete numerical solutions, that might be caused by e.g., the bisection method. This characteristic ensures robust and consistent calculations across various scenarios. Furthermore, the study by Koo et al. (2020) provided an independent evaluation of EQSAM's numerical stability, confirming it's free of numerical noise.

We have revised the sentence at line 36-38 to better reflect this by:
"… summer months. It was found that EQSAM4Clim accurately parameterises the gas/liquid/solid aerosol partitioning and associated aerosol water uptake sufficiently fast and free of numerical noise (Koo et al., 2020), which is true at all time scales."

*- with respect to the EQSAM4Clim-v12 being (sufficiently) accurate, it'd be helpful to also outline the cases/conditions when it is less accurate (e.g. specific aerosol compositions, regimes, and probably for very high RH, see Specific comments)*

We appreciate the suggestion to delineate the scenarios wherein EQSAM4Clim-v12 may exhibit diminished accuracy. Currently, the most notable area of concern relates to the presence of bi-sulfate in solution, ie., under conditions where there is an inadequate supply of cations to fully neutralize all sulfate (e.g., SOAS, Calnex and the WINTER campaign cases, Figure 7b). This has been already mentioned at line 203.

We have revised the sentence at line 203 to better reflect this by:
"… due to limitations of the $SO^{2-}$ / $HSO^-$ partitioning of the EQSAM4Clim version. Currently this is the weakest part and therefore the deviation in pH from E-AIM is largest. This will be subject for improvement in further updates."

*- Regarding high RH, the values as high as 98-99 and even 100% from weather prediction models can be in the meteorological inputs to CTMs. Such cases do not seem to be considered in this note. I think that a very clear recommendation should be made (actually it'd be very important) on the applicability of the parameterization*

We acknowledge the importance of considering high relative humidity (RH) conditions, including values nearing 100%, in atmospheric modeling. However, it's important to note that the evaluation of EQSAM4Clim-v12 under such extreme RH regimes is beyond the scope of this study. While the applicability of the proposed parameterization might be less of a concern, the accuracy under such conditions basically depends on the liquid water content calculations, which in turn depends on upon the coupling assumptions between EQSAM4Clim-v12 and meteorological inputs. The representation of aerosol-cloud interactions and the dynamic limitations of gaseous uptake (including water vapour) might be here of primarily of concern.

We have added the following at line 208 to better reflect this by:
"… pH values < 0.0. Note that the proposed parameterization does not show a limitation to very low or high RH values, according to the results shown. Also, extremely high RH values as high above 98-99 and even 100%, which might be a meteorological input to EQSAM4Clim-v12 through a NWP coupling, are not a limiting factor in general. Instead, the representation of aerosol-cloud interactions and the dynamic limitations of gaseous uptake (including water vapour) might be here of primarily of concern, though the evaluation of EQSAM4Clim-v12 under such extreme RH regimes is beyond the scope of this study."

*- it'd be useful to summarize for which of the cases the discrepancies between EQSAM4Clim-v12 and E-AIM are largest/smallest (more/less acid, more/less complex chemical system). And thus, how could this affect the results in CTMs with less complex chemical systems (e.g. missing some of base cations)?*

We have summarized the cases where the discrepancies between EQSAM4Clim-v12 and E-AIM are largest/smallest through Figure 7b. It's tricky, however, to provide an indication how this can affect the results in CTMs with less complex chemical systems, but likely it's not much. Overall, it largely depends on the complexity of the coupling of EQSAM4Clim-v12 with aqueous phase chemistry, emissions and meteorology and especially on how many mineral cations are considered as they tend to largely determine the pH values through their neutralization potential of anions and acids.

**Specific Comments**

*Section 2.2.1 lines 96-103: could you better explain where those "correction factors" come from; how the values were found*
Section 2.2.1 lines 96-103: The correction factors have been iteratively derived by comparing the diagnostic pH output of EQSAM4Clim-v12 with the reference pH computations of E-AIM for these five cases (using error minimizing on the log-scale) as noted on line 180ff.

*Is LWC_equilib (eq. 9a and l. 139) the same as AW?*

Line 139: LWC_equilib is the same as AW in case no other LWC source is considered. Depending on the 3D implementation, AW may also include LWC_noneq.

*146-147: could you outline the main implications of not using pH and H+ in EQSAM*

We appreciate the inquiry regarding the implications of not incorporating pH and H+ in EQSAM. It's important to note that within the framework of EQSAM, these properties serve purely as diagnostic tools and do not directly influence the underlying concept for its gas/aerosol partitioning. The model's formulation is based on a solution-independent single-solute coefficient (Metzger et al., 2012), which remains unaffected by variations in solute activity, including H+. While this approach diverges from conventional thermodynamic models, our analysis suggests that it remains valid within the uncertainty range presented by this study.

*151: Sounds a bit strange to recommend E-AIM for "accurate pH calculations" after it's been declared that EQSAM4Clim-v12 is "accurate/sufficiently accurate". Maybe to specify what calculations/applications/particular cases etc the authors mean here.*

Line 151: We recommend E-AIM for reference since it is much more explicit than EQSAM4Clim-v12. E-AIM also has been successfully applied to a wide range of applications. This is why we also use E-AIM for this box model evaluation. E-AIM is however not available nor suitable for 3D applications.

We have revised the sentence at line 152 to better reflect this by:
"… considered instead, as it has been successfully applied to a wide range of applications. E-AIM is, however, not available nor suitable for 3D applications."

*152-153: please list the known to the authors limitations for the parameterisation, the cases when it's not applicable*

Line 152-153: The parameterisation covers a wide range of atmospheric conditions, as shown in this study. The accompanying study of Rémy et al., 2024 (https://doi.org/10.5194/egusphere-2023-3072) expands this evaluation to the global scale. So far we are not aware of limitations for the current parameterisation.

We have changed the sentence at line 153 to reflect this by:
"… (see Sect. 3 and the accompanying study of Rémy et al. (2024))."

And we have added a final sentence at line 267 to reflect this by:
"… mandatory. The accompanying study (Rémy et al., 2024) expands this evaluation to the global scale."

*180-183: Explain more transparently the correction factors. Are those general or valid only for the considered cases? In the latter case, how should they be derived for new application cases?*

Line 180-183: The correction factors are most probably not only valid only for the considered cases, but could be valid potentially in general (within the EQSAM4clim-v12 framework) according to our 3D applications.

*190-191: and elsewhere: explain what should be expected at RH approaching 100%*

Line 190-191: We have added a final sentence at line 207 to reflect this by:

"In general, as RH approaches 100%, one can anticipate increasing pH values, primarily driven by the corresponding increase in liquid water content. This rise in pH results from the dilution of $H^+$ ions, leading to a reduction in acidity for a given $H^+$ concentration. However, the situation becomes more complex in the presence of soluble gases that form acids, as the dissolution of acids can dampen the increase in pH. Additionally, factors such as the presence of ammonia, clouds, or precipitation further complicate this picture. For a more in-depth discussion on these intriguing aspects, we direct interested readers to the accompanying studies by Rémy et al. (2024), and Williams et al. (to be submitted soon)."

*218-220: "different aerosol compositions…" - different from what? Or if different between them, what's the connection to the "most differing results"?*

Line 218-220: We refer to "different between them". The connection to the "most differing results" is the different EQSAM versions that are compared for the different campaign cases.

We have revised the sentence at line 223 to better reflect this by:
"… Additionally, comparing the different campaign cases shows …"

*264: what about less complex aerosol cases?*

Line 264: The less complex aerosol cases are addressed in line 223ff.
We have added a sentence at line 264 to complete this by:
"… ammonium compounds, while the less complex cases show a relatively larger scatter due to the limitations of the current $SO^{2-}$ / $HSO^-$ partitioning parameterization. Overall, EQSAM4Clim has a low cost with respect to the total CPU consumption …"

**Technical revisions**

*3: Probably should be "chemical weather prediction"*
3: Changed to "chemical weather prediction".

*7: (referred to as a version…)*
7: Changed accordingly.

*59: oxidation products of emissions from natural sources and …*
59: Changed accordingly.

*62: add i.e. or "namely before listing the cations*
62: Added "i.e.,"

*91: should be "dependent"*
91: Changed to "which depends on"

*106, 108: remove redundant ")"*
106, 108: Changed accordingly.

*117-118 and 144 - repetitive explanations for LWC0 and molality0 (already given on L. 109-110)*
117-118 and 144: Changed accordingly.

*156 : "which also was used" instead of "where this data has been used"*
156: Changed accordingly.

*159: in chemical transport models? (large scale)*
159: Deleted "applied in large-scale models".

*164: sorted by decreasing? Complexity*
159: Changed to "sorted by decreasing complexity".

*178-179: please add the upper range RH value (above 20% and up to XX % RH)*
178-179: Changed to "(between 20-90% RH)"

*185-186: repetition*
185-186: Deleted "as discussed in Pye et al. (2020)".

*200: Suggestion: These results show similar variability of AW content…*
200: Changed accordingly to "These results show similar variability of AW …".

*217, 223: What are those (a) and (b) for?*
217-223: "(a) and (b)" have been deleted (indeed not needed)

*217: I'd say "the pH results differ the most for Cabauw"*
217: Please note that we have used the data unfiltered (in contrast to Pye et al). The relatively large deviation in pH is limited here to a few spikes (erroneous data?) which might be ignored. For all the other data points the agreement is better than for the other campaign cases.

*218-220: the sentence seems incomplete. Probably it should be merged with the next one.*
218-220: Changed sentence to: "The Cabauw and Tianjin campaigns represent the most complex aerosol system with $SO_4^{2-}$ being fully neutralized (Sect. 3), since both locations are affected by anthropogenic precursors which undergo gas/aerosol partitioning."

*Section 3.2: I think it would make more sense to at least compare IFS with 3h averages from the box models to reduce at least a bit the inconsistency. Why are there several gaps in the box model results?*
In Section 3.2, we present a figure illustrating the application of the pH parameterization to IFS for the year 2019, while the box model results are shown for 2012/2013. It's important to clarify that the purpose of this figure is not to provide a quantitative comparison but rather to offer an initial demonstration of the parameterization's application to IFS outputs. The comparison spans different time periods (2012/2013 vs. 2019), and as such, quantitative assessments are not feasible. For a comprehensive evaluation of IFS performance, we direct interested readers to the companion study by Rémy et al., 2024 (https://doi.org/10.5194/egusphere-2023-3072), where a detailed assessment of IFS is conducted. Additionally, regarding the gaps in the box model results, we acknowledge these inconsistencies that are simply caused by a lack of data availability.

We have revised the sentence at line 246 to reflect this by:
"The implementation provides global pH values, whose impact on aerosol composition and PM2.5 is evaluated in the accompanying study of (Rémy et al., 2024)".

---

## Author Response (AR4)

**Authors' Response to comments of Anonymous Referee #1**

**General Comments**

*The paper presents another important evaluation for the efficient thermodynamic and chemical aerosol model EQSAM including some additional information on parameterizations and an application. This should be reflected more clearly in the title. EQSAM can be used as module in global and regional models or forecasting systems which should be mentioned explicitly in the introduction. The manuscript might be published after minor revision.*

We thank the anonymous reviewer for her/his positive response and appreciate the thorough review and constructive feedback. We have addressed all comments to enhance the clarity and quality of our manuscript. Our point-by-point response (in black) addresses all comments (in blue).

Title Clarification: We acknowledge your suggestion regarding reflecting the importance of the evaluation of the EQSAM model more explicitly in the title. We have revised the title to better emphasize the significance of the evaluation and its application in global and regional models or forecasting systems. New title: "*A computationally efficient parameterization of aerosol, cloud and precipitation pH for application at global and regional scale (EQSAM4Clim-v12)*".

Introduction Clarity: Thank you for highlighting the importance of explicitly mentioning EQSAM's applicability as a module in global and regional models or forecasting systems in the introduction.

We have revised the introduction accordingly at line 24ff: "*This makes EQSAM4Clim suited not only for climate simulations, but also* applicable to *air quality applications at the regional and global scale and ideal for high resolution Numerical Weather Predictions (NWP) …*"

**Specific Comments**

*Line 96: This sentence has to be improved for clarity. Does this mean a total factor of 100 in case D3? The use of 'N' together with 'XN' is also confusing, better use another letter (or 'Xn').*

Line 96: Yes, this means a total factor of 100 in case D3. We'll consider using 'YN' instead of 'N' and we have revised the sentence for clarity to:
"For cases outside this range ($XN \geq 0.9$ or $T \geq 293K$), XN needs to be scaled by 10 and multiplied by the factor YN given in Table 1, …"

*Line 159: "including E-AIM" should be inserted after "models" for better understanding the following.*

Line 159: We have inserted "including E-AIM" after "models" for better comprehension. The revised sentence reads:
"In Pye et al. (2020) five distinct cases were defined and used to evaluate the simulated pH of various thermodynamic models applied in large-scale models, including E-AIM."

*Lines 167-177: This can be expressed in a shorter way.*
Lines 167-177: We have condensed it to two lines (instead of three) for each bullet by replacing "2646 data points" by "N=2646", "The Netherlands" by "NL", "China" by "CN", and by deleting "composition including" and "reduced composition" for each bullet point.

*Line 234: Refer also to Fig 7b and say that V10 is worse here.*

Line 234: We'll refer to Fig 7b and specify that V10 performs worse in this scenario:
"Bi-sulfates are not always captured in the gas/liquid partitioning compared to cases which include semi-volatile compounds (Cabauw, Tianjin, WINTER), but v12 still outperforms v10 (comparing Fig. 7a and b)."

*Technical corrections*

*Line 10: Define acronym, line 44 is too late.*
Line 10: Changed to: … Extended Aerosol Inorganics Model (E-AIM) …
*Line 13: The acronym is already defined in abstract.*
Line 13: Changed to: EQSAM was developed …
*Line 16: Define acronym IFS, line 28 too late.*
Line 16: Changed to: … ECMWF's Integrated Forecasting System (IFS) …
*Line 25: Acronym already defined in abstract.*
Line 25: Changed to: … NWP …

*Line 17: Has the accompanying paper a reference? Or the references in lines 27, 243f? Define acronym.*
Line 17: We have added the missing references of the the accompanying paper Rémy et al. (2024) (https://egusphere.copernicus.org/preprints/2024/egusphere-2023-3072) and we have defined the IFS-COMPO acronym so that the sentence now reads:

"… implemented in the Integrated Forecasting System (IFS), with extensions to represent aerosols, trace and greenhouses gases, being called "IFS-COMPO" (also previously known as "C-IFS", Flemming et al. (2015)); see the accompanying paper (Rémy et al. (2024))."

*Line 74: Define also Z\* and H+\* here (missing for Eqn.5).*
Line 74: We have extended the sentence to include: "… (following the neutralization reaction order given by Table 3 of Metzger et al. (2016a)),  by using the effective hydrogen concentrations $H^{+,*}$ and $Z^*$ that are derived from Eqs. (1–8)."

*Line 84: Parentheses messed up.*
Line 84: Changed to: … "auto-dissociation of water $K_w$ [$mol^2$ /$kg^2$ ($H_2O$)]"."

*Line 91: Typo*
Line 91: Changed to: "depends".

*Lines 117 and 144: Remove the repetitions of the definitions in line 109f.*
Line 117, 144: We have removed the repetitions of the definitions.

*Line 166ff: This punctuation is confusing, use ',' instead of '-' between species and '+' instead of '/' as in caption of Fig. 2.*
Line 166ff: We have changed the punctuation accordingly.

*Figure 2 to 6: It would be better to use actual time as abscissa than the data point number (as Fig. 8). The additional title near the abscissa of panel d should be removed and might appear in caption.*
Figure 2-6: We prefer not to consider using the actual time as the abscissa for Figures 2 to 6, since the data gap is too large which will reduce and not enhance clarity. The gaps can be seen in Figure 8 for the Cabauw case, which shows the data too condensed to be clear for most of the other cases.

*Figure 8: Don't mess up different notations for time at abscissa (dd/mm/yy and mm.dd.yyyy.hh).*
Figure 8: We have changed the time notation for Figure 8.

**Authors' Response to comments of Anonymous Referee #2**

**Summary**

*The Technical Note presents an updated version of EQSAM, i.e. EQSAM4Clim-v12, which has been improved to simulate a more accurate pH of aerosol associated water (AW), and also in cloud water and precipitation. Accurate, numerically robust and efficient schemes to calculate the gas/aerosol partitioning, aerosol water and pH are indeed in large demand by regional and global chemical transport models (CTM) for both air quality assessments and forecasts. My recommendation is that the paper can be published in EGUsphere after minor revisions.*

We thank the anonymous reviewer for her/his positive response and appreciate the thorough review and very detailed feedback. We have addressed all comments to enhance the clarity and quality of our manuscript. Our point-by-point response (in black) addresses all comments (in blue).

**General Comments**

*I do not understand why it's stressed that the EQSAM4Clim-v12 is primarily meant for "chemical forecasting systems". It can indeed be implemented in any CTM used for Chemical Weather prediction and Air quality simulations, but also in climate models.*

We appreciate the clarification regarding the applicability of EQSAM4Clim-v12 beyond chemical forecasting systems. While the model is indeed versatile and can be implemented in various modeling frameworks, our emphasis on its use in chemical forecasting systems stems from its primary focus on providing predictions of atmospheric composition and pH levels. Nevertheless, we have changed the title and introduction to be more general.

New title: "*A computationally efficient parameterization of aerosol, cloud and precipitation pH for application at global and regional scale (EQSAM4Clim-v12)*".

Introduction changed at line 24ff: "*This makes EQSAM4Clim suited not only for climate simulations, but also* applicable to *air quality applications at the regional and global scale and ideal for high resolution Numerical Weather Predictions (NWP) …*"

*Then, I'd recommend to include in the relevant sections:*
*- brief explanation of the need/importance of modelling aerosol water pH*

Aerosol water pH plays a crucial role in the atmospheric chemistry of aerosols. It influences the aqueous uptake of gases such as sulfur dioxide or ammonia, which are key components affecting atmospheric composition and air quality. The acidity or alkalinity of aerosol water affects chemical reactions, including the formation and transformation of aerosol particles, as well as their interactions with other atmospheric constituents. Consequently, accurate modeling of aerosol water pH is essential for understanding and predicting processes related to atmospheric composition, aerosol-cloud interactions, and ultimately, their impact on climate and human health.

We therefore have added a short sentence about its impact at line 151:
"*… processes. The pH of aerosols controls their impact on climate and human health.*"

*- more clear outline of the improvements introduced with respect to the previous version and what particular processes and results (in what particular cases) they improved (was it only the factor XN, line 91?)*

We appreciate the request for a clearer outline of the improvements introduced in EQSAM4Clim-v12 compared to its previous version. Notably, EQSAM4Clim-v12 introduces a refined parameterization that separates aerosol, cloud, and precipitation pH, addressing a limitation of previous versions. This enhancement allows for more accurate representation of pH variations across different atmospheric conditions. The role of the XN correction factors can have a strong local impact in improving model performance as shown with Fig 7.

In the revised manuscript, we have added the following sentences at line 214:
"… study. Also note that both versions only differ by Eqs. (1-9e) with the results shown being sensitive to the Eq. (8) and the correction factors given in Table 1. Finally, note that what is most important for 3D applications is the fact that version 12 introduces a refined parameterization that separates the pH of aerosol, cloud and precipitation and addresses a limitation of previous versions through Eqs. (9a-9e). For a in-depth analysis we refer to the accompanying study of Rémy et al. (2024)."

*- what tests have been performed to check that the scheme is "free for numerical noise" (lines 4, 38, 57).*

We appreciate the inquiry regarding the numerical stability of the EQSAM4Clim-v12 scheme. As an analytical code, EQSAM inherently yields numerically stable results without introducing uncertainty through incomplete numerical solutions, that might be caused by e.g., the bisection method. This characteristic ensures robust and consistent calculations across various scenarios. Furthermore, the study by Koo et al. (2020) provided an independent evaluation of EQSAM's numerical stability, confirming it's free of numerical noise.

We have revised the sentence at line 36-38 to better reflect this by:
"… summer months. It was found that EQSAM4Clim accurately parameterises the gas/liquid/solid aerosol partitioning and associated aerosol water uptake sufficiently fast and free of numerical noise (Koo et al., 2020), which is true at all time scales."

*- with respect to the EQSAM4Clim-v12 being (sufficiently) accurate, it'd be helpful to also outline the cases/conditions when it is less accurate (e.g. specific aerosol compositions, regimes, and probably for very high RH, see Specific comments)*

We appreciate the suggestion to delineate the scenarios wherein EQSAM4Clim-v12 may exhibit diminished accuracy. Currently, the most notable area of concern relates to the presence of bi-sulfate in solution, ie., under conditions where there is an inadequate supply of cations to fully neutralize all sulfate (e.g., SOAS, Calnex and the WINTER campaign cases, Figure 7b). This has been already mentioned at line 203.

We have revised the sentence at line 203 to better reflect this by:
"… due to limitations of the $SO^{2-}$ / $HSO^-$ partitioning of the EQSAM4Clim version. Currently this is the weakest part and therefore the deviation in pH from E-AIM is largest. This will be subject for improvement in further updates."

*- Regarding high RH, the values as high as 98-99 and even 100% from weather prediction models can be in the meteorological inputs to CTMs. Such cases do not seem to be considered in this note. I think that a very clear recommendation should be made (actually it'd be very important) on the applicability of the parameterization*

We acknowledge the importance of considering high relative humidity (RH) conditions, including values nearing 100%, in atmospheric modeling. However, it's important to note that the evaluation of EQSAM4Clim-v12 under such extreme RH regimes is beyond the scope of this study. While the applicability of the proposed parameterization might be less of a concern, the accuracy under such conditions basically depends on the liquid water content calculations, which in turn depends on upon the coupling assumptions between EQSAM4Clim-v12 and meteorological inputs. The representation of aerosol-cloud interactions and the dynamic limitations of gaseous uptake (including water vapour) might be here of primarily of concern.

We have added the following at line 208 to better reflect this by:
"… pH values < 0.0. Note that the proposed parameterization does not show a limitation to very low or high RH values, according to the results shown. Also, extremely high RH values as high above 98-99 and even 100%, which might be a meteorological input to EQSAM4Clim-v12 through a NWP coupling, are not a limiting factor in general. Instead, the representation of aerosol-cloud interactions and the dynamic limitations of gaseous uptake (including water vapour) might be here of primarily of concern, though the evaluation of EQSAM4Clim-v12 under such extreme RH regimes is beyond the scope of this study."

*- it'd be useful to summarize for which of the cases the discrepancies between EQSAM4Clim-v12 and E-AIM are largest/smallest (more/less acid, more/less complex chemical system). And thus, how could this affect the results in CTMs with less complex chemical systems (e.g. missing some of base cations)?*

We have summarized the cases where the discrepancies between EQSAM4Clim-v12 and E-AIM are largest/smallest through Figure 7b. It's tricky, however, to provide an indication how this can affect the results in CTMs with less complex chemical systems, but likely it's not much. Overall, it largely depends on the complexity of the coupling of EQSAM4Clim-v12 with aqueous phase chemistry, emissions and meteorology and especially on how many mineral cations are considered as they tend to largely determine the pH values through their neutralization potential of anions and acids.

**Specific Comments**

*Section 2.2.1 lines 96-103: could you better explain where those "correction factors" come from; how the values were found*
Section 2.2.1 lines 96-103: The correction factors have been iteratively derived by comparing the diagnostic pH output of EQSAM4Clim-v12 with the reference pH computations of E-AIM for these five cases (using error minimizing on the log-scale) as noted on line 180ff.

*Is LWC_equilib (eq. 9a and l. 139) the same as AW?*
Line 139: LWC_equilib is the same as AW in case no other LWC source is considered. Depending on the 3D implementation, AW may also include LWC_noneq.

*146-147: could you outline the main implications of not using pH and H+ in EQSAM*
We appreciate the inquiry regarding the implications of not incorporating pH and H+ in EQSAM. It's important to note that within the framework of EQSAM, these properties serve purely as diagnostic tools and do not directly influence the underlying concept for its gas/aerosol partitioning. The model's formulation is based on a solution-independent single-solute coefficient (Metzger et al., 2012), which remains unaffected by variations in solute activity, including H+. While this approach diverges from conventional thermodynamic models, our analysis suggests that it remains valid within the uncertainty range presented by this study.

*151: Sounds a bit strange to recommend E-AIM for "accurate pH calculations" after it's been declared that EQSAM4Clim-v12 is "accurate/sufficiently accurate". Maybe to specify what calculations/applications/particular cases etc the authors mean here.*
Line 151: We recommend E-AIM for reference since it is much more explicit than EQSAM4Clim-v12. E-AIM also has been successfully applied to a wide range of applications. This is why we also use E-AIM for this box model evaluation. E-AIM is however not available nor suitable for 3D applications.

We have revised the sentence at line 152 to better reflect this by:
"… considered instead, as it has been successfully applied to a wide range of applications. E-AIM is, however, not available nor suitable for 3D applications."

*152-153: please list the known to the authors limitations for the parameterisation, the cases when it's not applicable*
Line 152-153: The parameterisation covers a wide range of atmospheric conditions, as shown in this study. The accompanying study of Rémy et al., 2024 (https://doi.org/10.5194/egusphere-2023-3072) expands this evaluation to the global scale. So far we are not aware of limitations for the current parameterisation.

We have changed the sentence at line 153 to reflect this by:
"… (see Sect. 3 and the accompanying study of Rémy et al. (2024))."

And we have added a final sentence at line 267 to reflect this by:
"… mandatory. The accompanying study (Rémy et al., 2024) expands this evaluation to the global scale."

*180-183: Explain more transparently the correction factors. Are those general or valid only for the considered cases? In the latter case, how should they be derived for new application cases?*
Line 180-183: The correction factors are most probably not only valid only for the considered cases, but could be valid potentially in general (within the EQSAM4clim-v12 framework) according to our 3D applications.

*190-191: and elsewhere: explain what should be expected at RH approaching 100%*
Line 190-191: We have added a final sentence at line 207 to reflect this by:
"In general, as RH approaches 100%, one can anticipate increasing pH values, primarily driven by the corresponding increase in liquid water content. This rise in pH results from the dilution of $H^+$ ions, leading to a reduction in acidity for a given $H^+$ concentration. However, the situation becomes more complex in the presence of soluble gases that form acids, as the dissolution of acids can dampen the increase in pH. Additionally, factors such as the presence of ammonia, clouds, or precipitation further complicate this picture. For a more in-depth discussion on these intriguing aspects, we direct interested readers to the accompanying studies by Rémy et al. (2024), and Williams et al. (to be submitted soon)."

*218-220: "different aerosol compositions…" - different from what? Or if different between them, what's the connection to the "most differing results"?*
Line 218-220: We refer to "different between them". The connection to the "most differing results" is the different EQSAM versions that are compared for the different campaign cases.

We have revised the sentence at line 223 to better reflect this by:
"… Additionally, comparing the different campaign cases shows …"

*264: what about less complex aerosol cases?*
Line 264: The less complex aerosol cases are addressed in line 223ff.
We have added a sentence at line 264 to complete this by:
"… ammonium compounds, while the less complex cases show a relatively larger scatter due to the limitations of the current $SO^{2-}$ / $HSO^-$ partitioning parameterization. Overall, EQSAM4Clim has a low cost with respect to the total CPU consumption …"

**Technical revisions**
*3: Probably should be "chemical weather prediction"*
3: Changed to "chemical weather prediction".

*7: (referred to as a version…)*
7: Changed accordingly.

*59: oxidation products of emissions from natural sources and ...*
59: Changed accordingly.

*62: add i.e. or "namely before listing the cations*
62: Added "i.e.,"

*91: should be "dependent"*
91: Changed to "which depends on"

*106, 108: remove redundant ")"*
106, 108: Changed accordingly.

*117-118 and 144 - repetitive explanations for LWC0 and molality0 (already given on L. 109-110)*
117-118 and 144: Changed accordingly.

*156 : "which also was used" instead of "where this data has been used"*
156: Changed accordingly.

*159: in chemical transport models? (large scale)*
159: Deleted "applied in large-scale models".

*164: sorted by decreasing? Complexity*
159: Changed to "sorted by decreasing complexity".

*178-179:  please add the upper range RH value (above 20% and up to XX % RH)*
178-179: Changed to "(between 20-90% RH)"

*185-186: repetition*
185-186: Deleted "as discussed in Pye et al. (2020)".

*200: Suggestion: These results show similar variability of AW content…*
200: Changed accordingly to "These results show similar variability of AW …".

*217, 223: What are those (a) and (b) for?*
217-223: "(a) and (b)" have been deleted (indeed not needed)

*217: I'd say "the pH results differ the most for Cabauw"*
217: Please note that we have used the data unfiltered (in contrast to Pye et al). The relatively large deviation in pH is limited here to a few spikes (erroneous data?) which might be ignored. For all the other data points the agreement is better than for the other campaign cases.

*218-220: the sentence seems incomplete. Probably it should be merged with the next one.*
218-220: Changed sentence to: "The Cabauw and Tianjin campaigns represent the most complex aerosol system with $SO_4^{2-}$ being fully neutralized (Sect. 3), since both locations are affected by anthropogenic precursors which undergo gas/aerosol partitioning."

*Section 3.2: I think it would make more sense to at least compare IFS with 3h averages from the box models to reduce at least a bit the inconsistency. Why are there several gaps in the box model results?*
In Section 3.2, we present a figure illustrating the application of the pH parameterization to IFS for the year 2019, while the box model results are shown for 2012/2013. It's important to clarify that the purpose of this figure is not to provide a quantitative comparison but rather to offer an initial demonstration of the parameterization's application to IFS outputs. The comparison spans different time periods (2012/2013 vs. 2019), and as such, quantitative assessments are not feasible. For a comprehensive evaluation of IFS performance, we direct interested readers to the companion study by Rémy et al., 2024 (https://doi.org/10.5194/egusphere-2023-3072), where a detailed assessment of IFS is conducted. Additionally, regarding the gaps in the box model results, we acknowledge these inconsistencies that are simply caused by a lack of data availability.

We have revised the sentence at line 246 to reflect this by:
"The implementation provides global pH values, whose impact on aerosol composition and PM2.5 is evaluated in the accompanying study of (Rémy et al., 2024)".

**Authors' Response to comments of the Executive Editor**

*Thank you for revising the manuscript. The reviewers agree that only minor issues remain to be addressed, especially regarding the figures (see Report 1) and the correction factors (see Report 2).*

We thank the Executive Editor and the anonymous reviewers for the thorough review and detailed feedback. To address the concerns regarding the clarity of the figures, we have improved them accordingly (i.e., corrected time abscissa for Fig. 2-6d, fixed truncated y-axis for Fig. 8 and more descriptive caption). Additionally, we show Figures 2-5 with a real time axis instead of data point number, Fig. 6 is kept as is but a version with real time axis is shown in a supplement. To further enhance the clarity and quality of our manuscript, we have addressed all other comments (in blue) regarding typesetting and additional comments.

*Eq. (1): Please typeset "tAnions" and "tCations" in roman font (non-italic using \mathrm).*
*Eq. (2): Chemical elements in roman font.*
*Eq. (3): Italic $K\_\mathrm{w}$, $T$, $T\_0$, $A\_T$, but roman K (Kelvin) and "where"*

All fixed accordingly.

*Eqs. (4), (8) etc.: Preferably use single-letter variables with subscripts rather than multi-letter variables to avoid confusion with products. Also consider the suggestion of referee 2 to use a more meaningful notation. For common multi-letter acronyms in other equations (e.g. LWC), please use roman font.*

To use a more meaningful notation and to avoid multi-letter variables for our correction factors (YN, XN), we have replaced throughout the text the chemical domain dependent factor, YN, by $K_D$. Accordingly, we have replaced the composition dependent factor, XN, which depends on the degree of neutralization of the given aerosol composition, by $F_N$.

*Eqs. (9): Roman pH, LWC, H, subscripts*

Fixed accordingly.

*In their AC, the authors clarified several issues and they added extra explanations/improved some unclear formulations in the paper which should improve the quality of the publication. There are still some suggestions before the paper can be recommended for final publication:*

*1. Regarding the recommendation to make a clearer outline of the improvements introduced in EQSAM4Clim-v12, the authors provide a good formulation in their AC (i.e. "EQSAM4Clim-v12 introduces a refined parameterization that separates aerosol, cloud, and precipitation pH, addressing a limitation of previous versions. This enhancement allows for more accurate representation of pH variations across different atmospheric conditions. The role of the XN correction factors can have a strong local impact in improving model performance").*
*Still, this has not been made transparent in the paper. First two sentences are only included in Results and Evaluation, while e.g. Abstract mentions only "aerosol acidity" (even though diagnosing pH of cloud and precipitation are among the main improvements).*

*The role of XN/YN correction factors are not mentioned either in Abstract or Conclusions (only in the technical description Sec. 2.2.1). I think it'd be relevant to mention calibration against E-AIM and introduction of the correction factors in Conclusion.*

We have revised the conclusion accordingly by: *"Using a range of diverse case studies we have performed box model calculations with the results being compared and calibrated against E-AIM, upon introducing domain and neutralization dependent correction factors, i.e., KD, and FN, respectively. The comparison against the E-AIM reference model calculations covers a range of seasons and scenarios ranging from forest measurements to maritime seaboard measurements."*

*2. Regarding XN scaling and YN correction factors, I wonder if it'd make it easier to follow the section if the authors could give a name to XN factor, reflecting its physical meaning (associated with non-ideal solution/neutralization??). It'd also help to emphasize XN scaling by 10 in footnotes to Table 1.*

*Further, the explanation of the origin of YN/XN ("The correction factors have been iteratively derived by comparing the diagnostic pH output of EQSAM4Clim-v12 with the reference pH computations of E-AIM for these five cases...") comes much later and in different section than where YN/XN are introduced (which means the given values appear kind of random).*

We have renamed the correction factors to be more meaningful (as mentioned already above), however, we retain from adding *scaling by 10* in footnotes to Table 1 as this would lead to a duplication of text. Also, we find the note about the origin of the correction factors in the results section more appropriate, as otherwise the different cases used would not have been introduced.

*3. Regarding the importance of modelling aerosol water pH, the authors gave a good explanation in their AC, whereas just one short general sentence ("The pH of aerosols controls their impact on climate and human health.") - which is OK, but the section with technical details of EQSAM4Clim-v12 doesn't seem to be the most appropriate place - perhaps could be moved to Introduction?*

Given the scope and style of this technical manuscript we prefer to keep it as is.